# Autophagy Deregulation in HIV-1-Infected Cells Increases Extracellular Vesicle Release and Contributes to TLR3 Activation

**DOI:** 10.3390/v16040643

**Published:** 2024-04-20

**Authors:** Catherine DeMarino, Maria Cowen, Anastasia Williams, Pooja Khatkar, Fardokht A. Abulwerdi, Lisa Henderson, Julia Denniss, Michelle L. Pleet, Delores R. Luttrell, Iosif Vaisman, Lance A. Liotta, Joseph Steiner, Stuart F. J. Le Grice, Avindra Nath, Fatah Kashanchi

**Affiliations:** 1Laboratory of Molecular Virology, School of Systems Biology, George Mason University, Discovery Hall Room 182, 10900 University Blvd., Manassas, VA 20110, USA; catherine.demarino@nih.gov (C.D.); mcowen4@gmu.edu (M.C.); awill57@gmu.edu (A.W.); pooja.khatkar@nih.gov (P.K.); michelle.pleet@nih.gov (M.L.P.); 2Section of Infections of the Nervous System, National Institute of Neurological Disorders and Stroke, National Institutes of Health, Bethesda, MD 20892, USA; lisa.henderson@nih.gov (L.H.); jdenniss97@yahoo.com (J.D.); delores.luttrell@nih.gov (D.R.L.); avindra.nath@nih.gov (A.N.); 3Basic Research Laboratory, National Cancer Institute, Frederick, MD 21702, USA; fardokht.abulwerdi@fda.hhs.gov (F.A.A.); legrices@mail.nih.gov (S.F.J.L.G.); 4Laboratory for Structural Bioinformatics, School of Systems Biology, George Mason University, Manassas, VA 20110, USA; ivaisman@gmu.edu; 5Center for Applied Proteomics and Molecular Medicine, George Mason University, Manassas, VA 20110, USA; lliotta@gmu.edu; 6Translational Neuroscience Center, National Institute of Neurological Disorders and Stroke, National Institutes of Health, Bethesda, MD 20892, USA; joe.steiner@nih.gov

**Keywords:** extracellular vesicles, human immunodeficiency virus-1 (HIV-1), combined antiretroviral therapy (cART), HIV-associated neurocognitive disorder (HAND), autophagy, central nervous system (CNS)

## Abstract

Human immunodeficiency virus type 1 (HIV-1) infection can result in HIV-associated neurocognitive disorder (HAND), a spectrum of disorders characterized by neurological impairment and chronic inflammation. Combined antiretroviral therapy (cART) has elicited a marked reduction in the number of individuals diagnosed with HAND. However, there is continual, low-level viral transcription due to the lack of a transcription inhibitor in cART regimens, which results in the accumulation of viral products within infected cells. To alleviate stress, infected cells can release accumulated products, such as TAR RNA, in extracellular vesicles (EVs), which can contribute to pathogenesis in neighboring cells. Here, we demonstrate that cART can contribute to autophagy deregulation in infected cells and increased EV release. The impact of EVs released from HIV-1 infected myeloid cells was found to contribute to CNS pathogenesis, potentially through EV-mediated TLR3 (Toll-like receptor 3) activation, suggesting the need for therapeutics to target this mechanism. Three HIV-1 TAR-binding compounds, 103FA, 111FA, and Ral HCl, were identified that recognize TAR RNA and reduce TLR activation. These data indicate that packaging of viral products into EVs, potentially exacerbated by antiretroviral therapeutics, may induce chronic inflammation of the CNS observed in cART-treated patients, and novel therapeutic strategies may be exploited to mitigate morbidity.

## 1. Introduction

As of 2021, 28.7 million individuals were estimated to be infected globally with human immunodeficiency virus type 1 (HIV-1), with 2.1 million new infection per year [1]. HIV-1 is a retrovirus that suppresses the immune system, promoting opportunistic infections and progression to acquired immunodeficiency syndrome (AIDS). The current gold standard of care is combination antiretroviral therapy (cART), which substantially decrease the viral loads in circulation by inhibiting nearly all stages of the virus cycle, from viral entry into the host cell to viral budding from the infected cell. Suppression of viral spread has allowed for the rise of long-term HIV-1 patients by increasing longevity and quality of life [2].

Antiretroviral drugs are classified into the following categories: non-nucleoside reverse transcriptase inhibitors (NNRTIs), nucleoside/nucleotide reverse transcriptase inhibitors (NRTIs), protease inhibitors (PIs), integrase strand transfer inhibitors (INSTIs), fusion inhibitors, CCR5 antagonists, CD4 post-attachment inhibitors, and pharmacokinetic (PK) enhancers [3,4]. Initial treatment generally includes pairing two NRTIs and either a PI, INSTI, or NNRTI [3]. While these drugs effectively suppress virus replication, the increased lifespan of long-term, chronically infected patients undergoing cART has led to the development of HIV-associated neurocognitive disorders (HAND). HAND is present in about 20 to 50% of infected individuals [5]. From mild to severe symptoms, HAND patients can progress through three stages: asymptomatic neurocognitive impairment (ANI), mild neurocognitive disorder (MND), and HIV-associated dementia (HAD). The symptoms of these stages involve neurophysiological impairment with minimal, mild, and severe decrease in everyday function, respectively [6]. Overall, typical clinical manifestations of HAND primarily include deficits in functions executed by the prefrontal cortex such as attention, memory, impulse control, and, to a lesser degree, motor skill dysfunction (balance and coordination), with the latter being more prominent in cART-treated individuals [7]. Specialized cART regimens have been optimized for patients exhibiting HAD symptoms, as these antiretrovirals most efficiently penetrate the blood–brain barrier (BBB). Specifically, these can include a combination of the PI darunavir, the NRTIs tenofovir and emtricitabine, and the PK enhancer ritonavir or cobicistat [3]. Despite their effectiveness in lowering viremia, current regimens lack the inclusion of an HIV-1 transcription inhibitor, thereby allowing for the continuous production of viral RNAs [8,9,10].

In response to the chemotactic signaling molecules CCL2 and CXCL12, infected monocytes migrate into the brain, differentiate into macrophages, and secrete a variety of chemokines, cytokines, neurotoxic host factors, and viral proteins during early infection [11]. Secretion of viral proteins can occur through a variety of mechanisms including their release as a result of cell death, the formation of defective viral particles, the shedding of the viral coat, and active/intended release [12]. Although there is suppression of the infectious virus, a lack of a transcription inhibitor allows for the production of short non-coding RNAs (as a result of non-processive RNA polymerase II transcription) and occasional full-length read-through transcripts that produce viral proteins. Several viral proteins and RNAs have been found in the cerebrospinal fluid (CSF) of HIV-1 infected individuals, potentially due to increased half-life through modifications inhibiting their reuptake and their encapsulation into extracellular vesicles (EVs) [12]. As a result of their increased half-life, viral products accumulate in the extracellular space within the central nervous system (CNS) when released from infected cells, traversing the BBB from the periphery to elicit neuroinflammation.

EVs are nanometer-sized, membrane-bound vesicles that are involved in intracellular communication through transportation of cellular cargo, such as nucleic acids, proteins, and lipids, to recipient cells. Studies have shown a change in the function of EVs associated with disease states, particularly those with CNS implications such as human T-lymphotropic virus [13,14], Epstein–Barr virus [15,16], and Kaposi’s sarcoma-associated herpesvirus [17]. We have previously investigated the role of EVs in HIV-1 infection, suggesting that EVs released from HIV-1-infected cells contain viral RNAs, specifically small non-coding RNAs such as trans-activating response element (TAR) [8,18,19], long non-coding RNAs such as the recently termed TAR-*gag* [20,21], and several viral proteins including Nef [8]. We have also shown that such EVs can contribute to the production of the pro-inflammatory cytokines IL-6 and TNF-β, thereby increasing the susceptibility of uninfected recipient cells to viral infection [19]. 

Autophagy is a dynamic cellular pathway that is responsible for the degradation/turnover of proteins and organelles, involving a series of coordinated events facilitated by numerous cellular proteins. The autophagy pathway, specifically secretory autophagy, intersects with pathways involved in EV biogenesis (for a complete review, see Pleet et al. [22]). In the context of disease, specifically viral infection, this process can be deregulated via viral proteins and can contribute to pathogenesis. New evidence suggests that viral proteins may not be the only catalyst for autophagy deregulation. Interestingly, the NRTI zidovudine (AZT) elicited mitochondrial DNA depletion in liver parenchymal cells, which in turn decreased pro-autophagy proteins and increased autophagy inhibitor activity. These effects resulted in increased accumulation of autophagosomes and proteins selectively degraded by autophagy [23]. Additional findings suggest these results are not limited to the liver, extending to cell types to include myocytes and adipocytes [24,25]. As AZT penetrates the BBB, albeit at low concentrations, these data may implicate cART drugs in the deregulation of autophagy in CNS cells. Further evidence has shown that Tenofovir and Emtricitabine can deregulate autophagy in circulating macrophages, measured by changes in the levels of autophagy proteins such as LC3-II and SQSTM1/p62 [26]. 

The interplay between autophagy and the release of EVs prompted an investigation into the role of antiretrovirals in the autophagy deregulation of HIV-1-infected cells and their impact on EV production. Herein, we demonstrate that antiretrovirals can impair autophagy, resulting in the accumulation of viral products and causing cellular stress. In previous studies, we showed that infected myeloid cells can alleviate the viral product burden intracellularly by secreting accumulating viral RNA and proteins in EVs. We now show that cART-mediated deregulation of the autophagy pathway increases the number of vesicles released from HIV-1-infected cells. Furthermore, we demonstrate that EVs from infected myeloid cells can activate the TLR3 contributing to inflammation during chronic infection. Finally, we present three compounds which, by binding HIV-1 TAR RNA, mitigate EV-mediated pathogenesis.

## 2. Materials and Methods

### 2.1. Cell Culture and Reagents

U937 (uninfected promonocyte), U1 (chronically HIV-1 infected U937 subculture), and CFF-STTG1 (uninfected astrocyte) cells were cultured in Roswell Park Memorial Institute (RPMI) 1640 media containing 10% fetal bovine serum (FBS), 1% penicillin/streptomycin and 1% L-glutamine (Quality Biological, Gaithersburg, MD, USA). Phorbol 12-Myristate 13-Acetate (PMA; 100 nM) was used to differentiate monocytes into monocyte-derived macrophages (MDMs) for 7 d [27,28]. U138-MG (astroglioma) cells were cultured in Eagle’s Minimum Essential Medium (EMEM) media with L-glutamine supplemented with 10% FBS and 1% L-glutamine. All the cells were incubated at 37 °C with 5% CO_2_ infusion. The U937 and U1 cells were treated with antiretroviral drugs, including darunavir (Cat# ARP-11447), ritonavir (Cat# ARP-4622), emtricitabine (Cat# ARP-10071), and tenofovir (Cat# ARP-10199), for 5 days at the indicated concentrations. The U1 cells (Cat# ARP-165), as well as the antiretroviral drugs, were provided by the AIDS Reagent Program (National Institutes of Health; Bethesda, MD, USA). The U937, CCF-STTG1, and U138 MG cells were obtained from the American Type Culture Collection (ATCC; Cat# CRL-1593.2, CRL-1718, and HTB-16 respectively). The cells were also treated for 5 days with varying concentrations of autophagy-inducing compounds (purchased from MedChemExpress; Monmouth Junction, NJ, USA), including: rapamycin (Cat# HY-10219), resveratrol (Cat# HY-16561), tamoxifen (Cat# HY-13757A), SAHA/Vorinostat (Cat# HY-10221), MG-132 (Cat# HY-13259), and valproic acid (Cat# HY-10585A). TAR-binding molecules have been previously described by Abulwerdi et al. [29] and are described in Appendix A.

### 2.2. Primary Macrophages

ScienCell Astrocytes (Cat# 1800) were grown in DMEM/F12 supplemented with 1× N2 (Cat# 17502048; ThermoFisher; Hillsboro, OR, USA), 10% FBS, and 1× antibiotic–antimycotic (ThermoFisher; Cat# 15240096). Neural stem cells were grown in DMEM/F12 supplemented with 5% FBS, 1× N2, 1× GlutaMAX (ThermoFisher; Cat# 35050079), and 1× antibiotic–antimycotic for three weeks and then maintained in DMEM/F12 supplemented with 10% FBS, 1× N2, 1× GlutaMAX, and 1× antibiotic–antimycotic. Human fetal astrocytes were obtained from human fetal brains of 10–14 weeks gestation, were grown for 4–6 passages, and were maintained in DMEM supplemented with 10% FBS, 1× GlutaMAX, and 1× antibiotic–antimycotic. The cells were seeded into 24-well plates coated with Geltrex (ThermoFisher; Cat# A1413202).

Healthy donors 1 (66F) and 2 (36M) were collected from fresh leukopaks obtained from the NIH Blood Bank. After collection, they were given approximately one week to differentiate into macrophages, with a half change of media (complete RMPI) occurring on the 3rd day of differentiation. Each donor was seeded into eight T-75 flasks, and half were infected with 89.6∆Env (BEI resources ARP-12486) HIV pseudotyped with 89.6 Envelope (BEI resources ARP-12485) at a MOI of 10 in unsupplemented RPMI. The other four were mock infected and exposed to 293T media. The cells were incubated overnight, and the next day, the cells were washed with Dulbecco’s phosphate-buffered saline (D-PBS), the media was replaced, and the cells were treated with antiretrovirals. The cells were retreated on day three, and the supernatants were collected on day five. The supernatants were cleared of cellular debris by centrifuging them at 300× *g* for 10 min, then EVs were isolated using 100,000 x g ultracentrifugation for 90 min. The EVs were used to treat ScienCell Astrocytes, neural stem cell-derived astrocytes, and human fetal astrocytes for 48 h. At 48 h post-exposure, the astrocytes were collected, and the RNA was isolated using Trizol (Invitrogen; Carlsbad, CA, USA) according to manufacturer’s protocol. The isolated RNA was resuspended in RNAse-free water and used to generate cDNA using the SuperScript III First-Strand Synthesis SuperMix for qRT-PCR (ThermoFisher; Cat# 11752250) according to the manufacturer’s protocol. The cDNA was analyzed using qPCR for IL-6 (F-GCAGAAAAGGCAAAGAATC, R-CTACATTTGCCGAAGAGC) and HPRT (F-GCTGACCTGCTGGATTACAT, R-GGTTTGCAGAGATTCAAAGAA). The PCR conditions were as follows: 95 °C for 20 s, followed by 40 cycles of 95 °C for 1 s and 60 °C for 20 s. The fold change was calculated using ∆∆CT, and a student’s *t*-test was used to determine statistical significance.

### 2.3. Preparation of Whole Cell Extracts

Uninfected and infected monocytes and macrophages were centrifuged at 15,000× *g* at room temperature for 5 min. The cell pellets were washed with 1× phosphate-buffered saline (PBS) without Ca^++^ and Mg^++^ and resuspended in lysis buffer comprising 120 mM NaCl, 50 mM Tris-HCl, pH 7.5, 50 mM NaF, 5 mM EDTA, 0.5% NP-40, 0.2 mM Na_3_VO_4_, 1 mM DTT, and 1 complete protease inhibitor cocktail tablet/50 mL (Roche Applied Science, Mannheim, Germany). The cell pellets were incubated on ice for 20 min and vortexed every 5 min. The whole cell lysates were separated from the cell debris via centrifugation at 10,000× *g* at 4 °C for 10 min. A Bradford assay (Bio-Rad; Hercules, CA, USA) was used to determine the protein concentration according to the manufacturer’s protocol.

### 2.4. Enrichment of EVs with Nanotrap Particles

Nanotrap (NT) particles (Ceres Nanosciences; Manassass, VA, USA) were used to enrich for EVs from low volumes. A three-equal-part slurry of NT80 particles (Ceres Nanosciences; Cat# CN1030), NT82 particles (Ceres; Cat# CN2010), and PBS was mixed, as described previously [8]. The supernatants were enriched for EVs with NT80/82 (30 µL particles/mL supernatant) and rotated overnight at 4 °C. The NT80/82 particles were pelleted at 15,000× *g* at room temperature for 5 min and washed with PBS.

### 2.5. Western Blot Analysis

Laemmli buffer (Tris-Glycine SDS and β-mercaptoethanol) was added to the whole-cell lysate samples and NT pellet. The samples were heated at 95 °C for 3 min, then loaded into 4–20% Tris–glycine gels (Invitrogen) with a Precision Plus Protein™ Standard (BioRad) and separated at 150 V. The gels were transferred onto Immobilon Polyvinylidene fluoride (PVDF) membranes (Millipore; Burlington, MA, USA) at 50 milliamps overnight. Five percent milk in PBS containing 0.1% Tween-20 (PBS-T) was used to block the protein membranes for 2 h at 4 °C. A primary antibody in PBS-T was added to the membranes prior to overnight incubation at 4 °C. The primary antibodies included α-SQSTM1/p62 (Cat# 5114; Cell Signaling Technology; Danvers, MA, USA), α-p24 (NIH AIDS Reagent Program; Cat# ARP-4121), α-CD63 (Cat# EXOAB-CD63A-1; Systems Biosciences; Palo Alto, CA, USA), MMP-3/10 (Cat# sc-374029; Santa Cruz Biotechnology; Dallas, TX, USA), IL-6 (Cell Signaling; Cat# 12153), and α-Actin (Cat# ab-49900; Abcam; Cambridge, UK). The membranes were washed three times with PBS-T (5 min/wash). Complementary HRP-conjugated secondary antibodies were added, followed by incubation for 2 h at 4 °C. The membranes were washed twice with PBS-T and once with PBS, with 5 min per wash. HRP luminescence was activated using Clarity Western ECL Substrate (Bio-Rad). The membranes were developed using the Molecular Imager ChemiDoc Touch system (Bio-Rad).

### 2.6. Ultracentrifugation for EV Isolation

Infected MDMs (1 × 10^6^ cells per mL) were cultured in 100 mL of RPMI media with 1% L-glutamine, 1% penicillin/streptomycin, and 10% exosome-free FBS per sample. The cells were pelleted via centrifugation at 2000× *g* for 5 min. The supernatant was then transferred into ultracentrifugation tubes, and the tubes were placed in a Ti70 (Beckman, Brea, CA, USA) rotor and centrifuged at 10,000× *g* for 30 min at 4 °C. Then, the supernatant was transferred to new tubes to be further centrifuged at 100,000× *g* for 90 min at 4 °C. The resulting pellet was then washed with PBS and pelleted at 100,000× *g* for 90 min at 4 °C. The pellets were resuspended in PBS and used to assess particle concentration and size via NTA.

### 2.7. EV Concentration Analysis

To assess for EV concentration and size of the cell-cultured EVs, Nanoparticle Tracking Analysis (NTA) was performed using ZetaView Z-NTA (Particle Metrix; software: ZetaView 8.04.02). The ZetaView was calibrated using 100 nm polystyrene nanostandard particles (Applied Microspheres, Leusden, the Netherlands) using sensitivity and minimum brightness settings of 65 and 20, respectively. The pre-acquisition parameters for each sample read 23 °C for temperature, 85 for sensitivity, 30 frames per second (fps) for frame rate, and 250 for shutter speed. One milliliter of diluted EV sample in deionized water was loaded in the cell. The average diameter size and concentration of the EVs were calculated and analyzed using ZetaView 8.04.02 and Microsoft Excel 2016 (v.15.0.4849.1003).

The primary macrophage extracellular vesicle sizes and concentrations were measured using MPRS (Spectradyne nCS1™, Spectradyne LLC, Torrance, CA, USA). The measurements were taken using TS-400 microfluidic cartridges. Ultracentrifuged samples were measured immediately after isolation with the addition of a 220 nm NIST bead spike-in population (1 × 10^9^/mL) for calibration. The post-acquisition analysis of Spectradyne nCS1 data was performed using RPSPASS software, which was developed in MATLAB (v 9.8.0.1323502 (R2022a, Mathworks Inc.; Natick, MA, USA)). The open-source code is available at: https://github.com/CBIIT//RPSPASS, accessed on 29 June 2019.

### 2.8. Cell Viability Assay

MDM and astrocyte cells were seeded at 5 × 10^4^ cells per well in a 96-well plate. Each sample was plated in triplicate, and cell-free media was used for background measurements. The cells were treated with their respective treatments and incubated for 5 d. CellTiter-Glo reagent Luminescence Viability kit (Promega; Madison, WI, USA) and the GLOMAX Multidetection System (Promega) were used to measure the luminescence and assess cell viability. Two plate readings were recorded, and the averages of both readings were calculated.

### 2.9. In Vitro Kinase Assay

Purified c-Src kinase (Cat# sc-19; Santa Cruz Biotechnology; 10 ng), TLR3 protein (Cat# NBP2-36506; Novus Biologicals; Centennial, CA, USA; 100 ng), TAR RNA (10–100 ng), and TAR inhibitor drugs (101FA, 102FA, 103FA, 104FA, 105FA, 106FA, 107FA, 108FA, 109FA, 110FA, 111FA, 112FA, 113FA, 115FA, 116FA, 120FA, Ral HCl; 1 µM) were washed with TNE_50_ buffer (Tris-HCl (pH 7.4), 100 mM NaCl, 0.1 mM EDTA) and kinase buffer, followed by a 1 h incubation period at 37 °C with γ-^32^P ATP. The samples were fractionated using SDS/PAGE through a 4–20% Tris–glycine gel, which was stained with Coomassie blue, destained in destain buffer, and dried for 2 h. The dried gels were developed using PhosphorImaging and quantified using ImageQuant Software (Molecular Dynamics; version 5.2).

### 2.10. Construction of Two-Dimensional and Three-Dimensional TAR RNA Structures

Two-dimensional (2D) and three-dimensional (3D) structures were generated using a published TAR RNA sequence (GGUCUCUCUGGUUAGACCAGAUCUGAGCCTGGGAGCUCUCUGGCUAACUAGGGAACCCACUGCUUAAGCCUCAAUAAAGCUUGCCUUGAGUGCUUC), as described previously [20]. The 2D structure was created using the RNAfold webserver. The minimum free energy (MFE) secondary structures for single RNA sequences was predicted using a loop-based energy model and the dynamic programming algorithm introduced by Zuker et al. [30]. RNA Composer was used to generate fully automated RNA 3D structures [31]. The secondary structure of TAR RNA in the Vienna format was retrieved from RNAFold [32] and was imported into the RNA Composer server. The TAR RNA 3D structure output was downloaded as a PDB file.

### 2.11. TAR-TLR3 Docking Analysis

The recognition of dsRNA by TLR3 is independent of its base sequence [33,34]; therefore, docking simulations between TAR RNA and TLR3 were performed using PATCHDOCK [35]. PATCHDOCK is a geometry-based molecular docking algorithm which divides the Connolly dot surface representation of the molecules into concave, convex, and flat patches. The complementary patches were then matched to generate candidate transformations. Each candidate transformation was further evaluated using a scoring function that considers both geometric fit and atomic desolvation energy [36]. Finally, root mean square deviation (RMSD) clustering was applied to the candidate solutions to discard redundant solutions. This algorithm has three major stages (i) Molecular Shape Representation, (ii) Surface Patch Matching, and (iii) Filtering and Scoring. The resulting docked conformations in PDB format were visualized using PyMol software (PyMOL-2.3.2_79-Win64-portable-py27.zip) [37]. The service used is available at: http://bioinfo3d.cs.tau.ac.il/PatchDock/ (accessed on 29 June 2019). The input parameters for the docking were the PDB coordinate file for the RNA and TLR3 molecules. Prior to TAR-TLR3 docking, the existing ligand was removed from the TLR3 PDB file from Protein Data Bank (3CIY) using PyMol (50), and a new PDB file was generated for 3CIY without the ligand. The Protein Data Bank-deposited structure 3CIY is a mouse Toll-like receptor 3 ectodomain (mTLR3-ECD) complexed with double-stranded RNA, and it shares a 78% sequence identity with the human Toll-like receptor 3 ectodomain (hTLR3-ECD) [34].

### 2.12. Densitometry Analysis

Densitometry analysis was performed using ImageJ software (version 1.54h). The densitometry data were normalized using a two-stop process to control for both exposure and loading. First, background measurements for each membrane were subtracted from the measurement of interest. Next, each protein band was normalized to the corresponding Actin. The normalized counts are represented as an increase or decrease relative to the untreated control (lane 1 set to 100%). Reduction trends were calculated by subtracting the normalized treated lane counts from the normalized control lane counts.

### 2.13. Real Time Quantitative Polymerase Chain Reaction (RT-qPCR)

Cells were harvested and washed once in 1× PBS free of calcium and magnesium. The pellets were then resuspended in 50 µL of 1× PBS free of calcium and magnesium. EVs were isolated using Nanotraps as described above. The total RNA was isolated through the use of Trizol Reagent (Invitrogen) according to the manufacturer’s protocol. cDNA was generated using GoScript Reverse Transcription Systems (Promega, Madison, WI, USA) with an envelope reverse sequence of 5′-TGG GAT AAG GGT CTG AAA CG-3′, a TAR reverse sequence of 5′-CAA CAG ACG GGC ACA CAC TAC-3′, and a TAR-*gag* reverse sequence of 5’-GCT GGT AGG GCT ATA CAT TCT TAC-3’. These were compared to quantitative standards that were created from serial dilutions of a CEM T-cell line that contained a single copy of HIV-1 LAV pro-virus per cell (8E5 cells).

For viral RNA, standards (2 µL) or cDNA samples (2 µL) were plated in a master mix (18 µL) composed of IQ Supermix (Bio-Rad, Hercules, CA, USA), TAR forward primer (5′-GGT CTC TCT GGT TAG ACC AGA TCT G-3′), TAR reverse primer (5′-CAA CAG ACG GGC ACA CAC TAC-3′), and TAR probe (5′56-FAM-AG CCT CAA TAA AGC TTG CCT TGA GTG CTT C-36-TAMSp-3′). The BioRad CFX96 Real Time System was used for RT-qPCR with the following conditions: a single 2 min cycle at 95 °C, followed by 41 cycles with 15 s at 95 °C and 40 s at 60 °C.

Each reaction was performed in triplicate, and quantification was determined by comparing the cycle threshold (Ct) values to the 8E5 standard curve in the BioRad CFX Manager Software (version 3.0). The analysis of generated raw data was performed in Microsoft^®^ Excel^®^.

### 2.14. Statistical Analysis

Standard deviations (SD) were used to assess for sample variance in all quantitative experiments via Microsoft Excel. A Student’s *t*-test was used to assess statistical significance or *p*-values. The *p*-values were considered * significant when 0.01 < *p* < 0.05, ** significant when 0.001 < *p* < 0.01, and *** significant when *p* < 0.001.

## 3. Results

### 3.1. Antiretroviral Drugs Alter EV Release from HIV-1-Infected Monocytes and Macrophages

Previously, we demonstrated that EVs released from HIV-1-infected monocytes carry viral proteins and RNAs despite cART [8]. To explore the effects of cART drugs on infected myeloid cell EVs, U1 and U937 cells were treated for 5 days with darunavir, ritonavir, emtricitabine, tenofovir, or a combination thereof, and the resulting supernatants were examined via Nanotracking analysis (NTA; Figure 1). The data in Figure 1A demonstrate that ritonavir (lane 3), tenofovir (lane 6), and to a lesser extent darunavir (lane 2) alone caused a statistically significant increase in EV release from the HIV-1-infected cells (58%, 31%, and 11% increase, respectively). Furthermore, when these cART drugs were used in combination, there was a 45% increase in EV release (lane 7). Conversely, in the absence of HIV-1 infection, darunavir (lane 2), ritonavir (lane 3), and tenofovir (lane 6) alone decreased EV production (Figure 1B). However, similar to U1, treatment of U937 cells with a combination of drugs (lane 7) increased EV release by 69%.

Data in Figure 1C show an increase in the median diameter size of the EVs released from HIV-1-infected monocytes following treatment with every tested cART drug (lanes 2–7). Conversely, in the treatment of uninfected U937 cells, cART did not cause a significant change in the mean diameter of the released EVs (Figure 1D). An analysis of the mean diameter size confirmed these findings, as the EVs released from the HIV-1-infected monocytes post-cART showed an increase in mean size, while the EVs released from uninfected monocytes post-cART remained the same (Appendix A). Furthermore, the peak (mode) diameter size did not change, regardless of HIV-1 infection (Appendix A). These data indicate that the increase in EV concentration may potentially be due to an increase in the release of large vesicles such as secreted autophagosomes.

Previous findings have shown that cART drugs can potentially interfere with the autophagy pathway in several different cell types [23,24,25,26]. To explore the effects of cART on autophagy in monocytes, U1 (HIV-1-infected monocytes) and U937 (uninfected monocytes) cells were treated for 5 days with darunavir (PI), ritonavir (a pharmacokinetic enhancer), emtricitabine (NRTI), tenofovir (NRTI), or a combination thereof and analyzed via Western blot. Changes in intracellular p62 (Sequestosome 1; SQSTM1), an autophagy marker protein that functions as an adaptor protein involved in the shuttling of cargo to the autophagosome, were assessed to determine whether the autophagosomes successfully fused with lysosomes to facilitate degradation of the intended cargo via hydrolases. As p62 is involved in sequestering the cargo to the autophagosome, there is selective degradation of p62 during normal autophagic flux. Therefore, it follows that an increase in p62 levels is indicative of autophagic impairment. The data in Figure 2A show an increase in the intracellular p62 levels when HIV-1 infected monocytes were treated with antiretrovirals, suggesting an impairment in autophagy-mediated protein degradation (top panel, lanes 2–7).

The most significant of these was observed for the darunavir/ritonavir treatment, which resulted in an increase in p62. In contrast, treatment of uninfected U937 cells with ritonavir alone (lane 3), tenofovir alone (lane 5), and a combination of all four drugs (lane 7) caused an increase in intracellular p62 levels (Figure 2B). The most significant increase was observed in the cells treated with ritonavir alone (lane 3), which may suggest an unintended consequence of inhibiting CYP3A4, the cellular enzyme target of ritonavir, which is involved in metabolism. Surprisingly, treatment with darunavir alone (lane 2) and a combination of darunavir and ritonavir (Figure 2B; lane 4) caused autophagy activation in uninfected monocytes, as indicated by a decreased p62 levels. Interestingly, darunavir treatment of infected monocytes, either alone or in combination with ritonavir (Figure 2A; lanes 2 and 4, respectively), induced cleavage of p62, resulting in a lower molecular weight and potentially defective forms. However, the same cleavage was not observed in uninfected monocytes (Figure 2B), suggesting viral involvement in this process. To verify the efficacy of the cART drugs, the levels of HIV-1 Gag p24 were also analyzed, along with HIV-1 viral RNA (TAR, TAR-*gag*, and *env*; Appendix A). Monotherapy resulted in an incomplete suppression of the virus (Figure 2A; top panel, lanes 2–6). Alternatively, when all four drugs (darunavir, ritonavir, emtricitabine, and tenofovir) were used in combination, there was a distinct reduction in the p24 levels, indicating successful viral suppression (Figure 2A; lane 7). In summary, these data suggest that cART drugs, particularly for the treatment of individuals with CNS infection, may deregulate autophagy in infected monocytes.

Given the interplay between the autophagy and EV pathways, we reasoned that cART-mediated inhibition of the autophagy pathway could potentially cause an increase in the release of EVs from HIV-1 infected cells, thereby contributing to viral pathogenesis. To test this notion, the supernatants from cART-treated U1 cells were enriched for EVs using Nanotrap particles and analyzed via Western blot. The results in the lower panel of Figure 2A show an increase in p62 levels in the conditioned culture media from cells treated with cART drugs either alone or in combination (lanes 2–7), indicative of an increase in secreted autophagosomes. Furthermore, treatment with cART drugs also elicited an increase in the levels of CD63, a well-characterized EV marker (lanes 2–7). Interestingly, the most significant increase in the measured proteins was noted in the cells treated with all four drugs (lane 7), which exhibited an increase in p62 and CD63. Conversely, decreased p62 and CD63 levels were observed in the supernatants of uninfected monocytes treated with cART drugs (Figure 2B; lanes 2, 4–7) with the exception of those treated with ritonavir alone (lane 3), in line with the observed autophagy inhibition as measured by intracellular p62 levels. Taken together, these findings suggest that cART impairs autophagy turnover in HIV-1-infected monocytes, which in turn may cause an accumulation of cargo within the cell. The accumulation of non-degraded cargo can cause cellular stress, resulting in increased release of secreted autophagosomes and other CD63^+^ EVs to alleviate the burden.

Given that HIV-1 infected macrophages are one of the primary cell types that contribute to CNS inflammation in HIV-1-infected individuals, we next sought to investigate whether administration of cART could cause the same autophagy impairment phenotype in infected macrophages. HIV-1-infected U1 and uninfected U937 cells were differentiated into macrophages and treated with the cART drugs previously described. To verify the efficacy of the cART treatment, viral RNA levels were assessed using RT-qPCR for viral TAR, TAR-*gag*, and *env* (Appendix A). Following a 5-day incubation period, the supernatants were analyzed using ZetaView NTA for size and concentration (Figure 3). Similar to the data shown in Figure 2A, treatment with darunavir (lane 2), ritonavir (lane 3), and tenofovir (lane 6) alone caused a significant increase in the number of EVs released from the infected macrophages (Figure 3A; 61%, 68%, and 67%, respectively). Furthermore, the combination treatments (lanes 4 and 7) also caused a significant increase in EV release (57% and 71%, respectively). A similar trend was observed when the same treatments were applied to uninfected macrophages, i.e., all the tested cART drugs elicited an increased release of EVs compared to the untreated control (Figure 3B). These findings are distinctively different from those observed for monocytes (Figure 1B), suggesting differential responses to cART drugs as a result of myeloid differentiation. Furthermore, when the median EV diameter was analyzed, the HIV-1 infected (Figure 3C) and uninfected (Figure 3D) macrophage EVs showed no change in response to cART treatment, which may imply an overall increase in all secreted EV populations, including both smaller vesicles such as exosomes and larger vesicles such as secreted autophagosomes. These findings were confirmed by an analysis of both the mean (Appendix A) and peak (Appendix A) diameter sizes. Taken together, these data suggest that cART regimens including a combination of darunavir, ritonavir, tenofovir, and emtricitabine can increase the release of EVs from HIV-1-infected monocytes and macrophages. However, the data may imply that the stage of differentiation may play a role in the type of EV that is increased with cART, contributing to increased production of larger vesicles from monocytes vs. increased released of all EV types in macrophages.

### 3.2. cART-Mediated Changes in Autophagy Contribute to HIV-1 Pathogenesis

As inhibiting the autophagy pathway resulted in an increase in EV release, we hypothesized that pharmacological induction of autophagy could potentially mitigate the increased EV release elicited by cART. There are several FDA-approved drugs for the treatment of various diseases that have been shown to induce autophagy [38,39,40,41]. Thus, HIV-1-infected U1 and uninfected U937 cells were differentiated into macrophages using PMA and were treated with cART as previously described. Additionally, macrophages were treated with autophagy-inducing compounds (rapamycin, resveratrol, tamoxifen, SAHA, MG-132, and valproic acid) at optimized concentrations as determined using a cell viability assay (Appendix A). Induction of autophagy was associated with a decrease in the number of EVs released from the HIV-1-infected cells (Figure 4A). Among the tested compounds, resveratrol (lane 3), tamoxifen (lane 4), SAHA (lane 5), valproic acid (lane 7), and to a lesser extent, rapamycin (lane 2), elicited decreased release of EVs from HIV-1-infected macrophages. Interestingly, the addition of autophagy-inducing compounds caused only a slight reduction in the number of EVs released from the uninfected cells (Figure 4B). Furthermore, treatment with autophagy inducers did not significantly change the median diameter size of the EVs from infected (Figure 4C) and uninfected (Figure 4D) macrophages, nor did they elicit a change in the peak diameter size or mean diameter size (Appendix A). Taken together, these data suggest that the induction of autophagy can specifically decrease the number of EVs from HIV-1-infected cells, with only a slight reduction in EVs from uninfected macrophages, which can contribute to normal cellular communication.

To confirm the effects of the antiretrovirals on autophagy, primary macrophages were infected with HIV-1 for 3 days. Following infection, the cells were treated with monotherapy or a combination of antiretrovirals (every 48 h) and transduced with a baculovirus tandem RFP-GFP LC3B sensor with hydroxychloroquine as a positive control. Cells were imaged using a confocal microscope at 24, 48, and 96 h post-treatment/transduction (Figure 5). All the combination therapies elicited a reduction in autophagosome fusion in both donors at all timepoints. Interestingly, the effects of the antiretrovirals on autophagy were seen at 24 h post-treatment, while the positive control (Chloroquine) did not inhibit autophagy until much later.

As HAND poses a threat to the quality of life of adequately suppressed HIV-1 patients, we next wanted to evaluate the role EVs had from infected macrophages on recipient CNS cells, specifically astrocytes. We have previously shown that EVs released from infected cells can cause an increase in the production of proinflammatory cytokines in uninfected recipient cells [19], and that these EVs can be found within the CSF of individuals infected with HIV-1 [8]. We therefore hypothesized that treating cells with EVs from infected macrophage-derived monocytes could elicit changes in astrocytes. To test this, U1 HIV-1-infected monocytes were differentiated into macrophages using PMA and treated twice with cART (darunavir, ritonavir, tenofovir, and emtricitabine) for 5 d. Donor cell viability was assessed to determine the optimal drug concentrations (Appendix A). Following incubation, total EVs were isolated using ultracentrifugation, and the EV concentration was normalized to control numbers using NTA (Appendix A). These EVs were used to treat two astrocyte cell lines, CCF-STTG1 and U138, at a 1:10,000 ratio of cell/EVs for 24, 48, 72, and 96 h. A sodium butyrate treatment was used as a positive control for inducing inflammatory signals [42]. Post-incubation, the astrocyte supernatants were analyzed for secreted inflammatory markers via Western blot analysis. The results presented in Figure 6A show an early strong secretion of IL-6 from CCF-STTG1 treated with EVs released from HIV-1-infected MDMs treated with cART at the 24 h time point, which sustained high levels of secretion (Appendix A) through 96 h. Interestingly, the sodium butyrate treatment did not induce IL-6 secretion from the astrocytes until after 48 h. An inflammatory marker, MMP3, showed a slight increase (Appendix A) in secretion induced by HIV-1 MDM EVs at 24 h but then remained at relatively consistent secretion levels compared to the untreated control. The EV and sodium butyrate treatments of U138 astrocytes had similar effects (Figure 6B) to those shown in Figure 6A.

We next collected EVs from HIV-1-infected primary macrophages treated twice with cART over 5 days. The concentration of these EVs was assessed using MPRS (Appendix A). Equal numbers of EVs were incubated with the astrocytes, and the IL-6 transcripts were assessed via qPCR. Similar to the U1 MDMs, there was an increase in IL-6 in the astrocytes treated with EVs from HIV-1-infected primary macrophages (Appendix A). Taken together, the EVs released from HIV-1-infected MDMs treated with antiretrovirals can induce IL-6 secretion from astrocytes, which may potentially contribute to neuroinflammation.

### 3.3. Small-Molecule TAR-Binding Compounds Inhibit TLR3 Activation and Mitigate EV-Mediated Effects on Astrocytes

Our previous studies have demonstrated that EVs released from HIV-1-infected cells contain TAR RNA, a 59-nucleotide, small, non-coding RNA produced in infected cells as a result of non-processive transcription [8,18,19]. The presence of TAR RNA within EVs has been confirmed in patient biofluids, including CSF, despite suppressive antiretroviral therapy [8]. Additional studies by our lab showed that EV-encapsulated TAR RNA can bind Toll-like Receptor 3 (TLR3), a receptor of the innate immune system involved in the sensing of double-stranded RNA, in recipient myeloid cells to elicit an increase in the production of proinflammatory cytokines through activation of the NF-κB pathway [19]. As TLR3 is expressed in numerous cell types, including several within the CNS such as glia (microglia, astrocytes, and oligodendrocytes) and neurons [43], we therefore questioned whether EV-mediated TLR3 activation could be responsible for the increase in IL-6 and thereby play a role in the chronic neuroinflammation observed in long-term, cART-treated patients [44,45,46].

In order to confirm TLR3 activation by TAR RNA, we utilized a functional in vitro kinase assay in which wildtype TAR (WT TAR), mutated TAR D (a mutant TAR missing most of the stem-loop structure), and mutated TAR TM26 (a TAR mutant with base substitutions in the Tat-binding region and within the loop structure) were compared to control IgG and Poly I:C. for Src kinase phosphorylation of TLR3. The data in Figure 7A show phosphorylation of TLR3 by Src kinase upon the addition of WT TAR RNA and with TAR mutant TM26 (Figure 7A; lane 4 and 6). This activation of Src is not observed when the stem-loop structure of TAR is modified (as seen with TAR D; Figure 7A; lane5). This indicates that the stem-loop structure of WT TAR is able to dimerize TLR3, allowing for phosphorylation by Src kinase. These results were quantitated using a densitometry analysis (Figure 7B), showing an approximately 55% increase in the phosphorylation of TLR3 in the presence of WT TAR RNA.

To inhibit this phosphorylation and subsequent activation, we utilized a panel of small molecules (Appendix A) that have previously been found to bind HIV-1 TAR RNA [29] and screened them for their ability to inhibit TLR3 activation using an in vitro kinase assay. The data shown in Figure 7C show decreases in the phosphorylation of TLR3 by Src kinase in 103FA, 111FA, and Ral HCl-treated immunoprecipitation complexes. The structures of the three candidates, 103FA, 111FA, and Ral HCl, are shown in Figure 7D.

These molecules have been shown to bind at TAR RNA residues U23, C24, C39, U40, and U41 via TOCSY NMR (54). We next performed docking experiments using TAR RNA and the TLR3 receptor to determine the binding residues involved in the interaction between these two molecules. TLR3 recognizes dsRNA, a molecular signature of most viruses, and triggers downstream signaling cascades that result in inflammatory responses to prevent viral spread. TLR3 ectodomains (ECD) dimerize upon binding oligonucleotides of at least 40–50 nt in length, the minimal length required for signal transduction, allowing dimerization of the Toll IL-1 receptor (TIR) domain and subsequent activation of signaling cascades [33,47,48]. A 96 nucleotide TAR RNA was used for docking, which forms a stem-loop structure, making it an ideal candidate as a TLR3 ligand. Using PATCHDOCK, the top 10 best solutions, as determined by their geometric shape complementarity score, interface area of the complex, and Atomic Contact Energy (ACE), were obtained (Table 1). The solution with the highest geometric shape complementarity score, the highest interface area, and the lowest ACE value was considered to be an ideally docked complex. Figure 7E shows the top-ranked solution visualized using PyMol (50).

The top candidate hits indicated that TAR RNA nucleotide residues U6, C7, G52, and G53 interact with TLR3 (Figure 7E). More specifically, TAR RNA residues U6 and G52 interact with the C-terminus residue His539, a well-described TLR3 activation residue [34,49,50,51], on two different monomeric units of TLR3, thereby causing dimerization of the ectodomains and initiation of the signaling cascade (Appendix A). Additionally, the data shown in Appendix A indicate that another TLR3 C-terminus residue, Asn515, a known RNA-binding residue that has not been shown to be required for TLR3 activation, also binds TAR RNA [34]. Along the same line, the TLR3 residues Asn541 and Leu 595 were found to interact with TAR RNA at the C7 and G53 positions, respectively (Appendix A). Asn541, a residue necessary for TLR3 activation (47, 71–73), showed interactions only at one monomeric unit. The interaction of TAR RNA with additional non-activating residues could potentially lend stability to the interaction, prolonging the induction of the signaling pathway.

Taken together, the molecular docking results suggest that TAR RNA may form a closed loop with TLR3, which coordinates and facilitates the dimerization of two TLR3 ectodomains. The TAR RNA nucleotides bound by the small molecules are independent of those nucleotides which bind TLR3 (Figure 7D); therefore, inhibition of TLR3 activation is unlikely to occur via direct competition. Instead, the interaction of TAR RNA with TAR inhibitors may elicit a change in the conformation of the RNA, rendering the RNA unrecognizable to TLR3 or potentially interfering with the dimerization of TLR3, preventing the formation of a closed loop.

Given that the EVs released from HIV-1-infected MDMs treated with cART elicited an inflammatory response in astrocytes, and that TAR-binding compounds lowered TLR activation induced by double-stranded TAR RNA, we reasoned that inhibiting TAR RNA-mediated TLR3 activation could potentially mitigate the inflammatory response in recipient astrocytes. To this end, HIV-1-infected U1 cells were differentiated into macrophages using PMA, treated with cART twice, and incubated for 5 d. Following incubation, EVs from uninfected and infected MDMs (U937 MDM and U1 MDM, respectively) were isolated using ultracentrifugation and normalized using NTA for EV concentration (Figure 3A,B). The median size similarities were also confirmed via NTA (Appendix A). Normalized EV preps at a ratio of 1:10,000 (cell:EV) and a titration of candidate TAR-binding compounds (1, 10, and 100 nM) were used to treat recipient, uninfected astrocytes (CCF-STTG1) and were incubated for 3 d. The data presented in Figure 8 confirm that EVs released from HIV-1 MDMs induced secretion of IL-6 from the astrocytes (lane 3), while the EVs released from the uninfected monocytes did not induce proinflammatory cytokine secretion (lane 2) compared to the untreated control (lane 1). The presence of three candidate TAR-binding compounds induced a dose-dependent decrease in secreted proinflammatory cytokine, IL-6, indicating that the TAR inhibitors 103FA, 111FA, and Ral HCl can lower subsequent proinflammatory cytokine production via binding of TAR RNA. Taken together, these data imply that TAR-binding compounds could potentially be used to alleviate HIV-1 EV-mediated chronic inflammation.

## 4. Discussion

We and others have previously shown that, despite effective viral suppression by cART, viral reservoirs, including those in the CNS, are transcriptionally active, resulting in the production of short non-coding and full-length genomic RNAs via non-processive and basal transcription, respectively. This is supported by studies reporting the persistence of approximately 1 × 10^3^ copies of cell-associated RNA in both circulating CD4^+^ T-cells [9,10] and myeloid cells from several brain regions [52]. Furthermore, we have found that these RNAs and their respective proteins can be packaged into EVs and released from the cell to elicit changes in uninfected recipient cells [8,18,19]. In line with this, we have shown here that cART can increase the release of EVs containing viral RNAs and proteins through deregulation of the autophagy pathway in HIV-1-infected myeloid cells, as measured by an accumulation of both p62 within the cell and the presence of secreted autophagosomes in the extracellular space (Figure 1 and Figure 2). The presence of secreted autophagosomes in the extracellular space potentially suggests that the cART-mediated inhibition of autophagy occurs at the level of lysosome fusion. The inhibition of degradation via autophagy results in a disruption of cellular turnover, which, in turn, can promote the accumulation of cell-associated viral RNAs and proteins in infected myeloid cells. An abundance of infected macrophages in the brain is correlated with HAND neuropathogenesis [53,54]. The data presented here suggest that increased EV release following infected macrophage migration into the brain could potentially play a role in HIV-1 neuropathogenesis. Interestingly, this mechanism may be dependent on the stage of differentiation of myeloid cells, as cART caused an increase in the secretion of larger vesicles such as autophagosomes from monocytes (Figure 1), whereas the production of all sizes of vesicles was increased in MDMs (Figure 3).

Autophagy, specifically mitophagy (a form of selective mitochondrial autophagy), is essential for basal turnover in cells and is of particular importance in maintaining neuronal homeostasis. The disruption of autophagy and/or mitophagy has been implicated in many chronic neurodegenerative diseases, such as Alzheimer’s disease, Parkinson’s disease, multiple sclerosis, amyotrophic lateral sclerosis, and transmissible spongiform encephalopathies (i.e., prion diseases) [55,56,57], suggesting the potential for elevated EV release. Along these lines, EVs have been shown to contribute to the propagation of these diseases via the transfer of various cargos, including misfolded proteins, from diseased to healthy cells [58,59,60,61,62,63,64,65,66,67]. It follows that the induction of autophagy via the inducers used here could constitute therapeutics to prevent or slow the progression of neurodegenerative diseases beyond HIV-1 neurological pathologies. Given the pivotal role of basal autophagy, including selective autophagy, in cellular homeostasis and quality control, it is not surprising that excessive reduction in these processes can elicit and contribute to disease. It follows that viruses, including HIV-1, have evolved ways to either circumvent the degradation of viral proteins through these processes, and that drugs used for treating infection could exacerbate these effects. Considering the many forms of autophagy and the diverse changes in these processes regarding disease, further research is needed to develop more specific modulators of autophagy to allow for precise targeting of the disease mechanism.

The emergence of extracellular vesicles as a mediator of disease has led to an increase in research into potential therapeutics to combat EV-mediate pathogenesis via various mechanisms. Our previous findings showed that tetracycline-class antibiotics can alter the endosomal sorting complexes required for the transport (ESCRT) machinery of myeloid cells to modulate EVs [8,68]. As both the ESCRT and autophagy pathways are involved in EV biogenesis, we have, in the current study, utilized a panel of autophagy-inducing compounds to promote the degradation of cell-associated cargo and limit the number of EVs released from infected cells. Our findings show that resveratrol (an mTOR inhibitor), tamoxifen (a Beclin-1 antagonist), SAHA (an mTOR inhibitor), and valproic acid (a histone deacetylase (HDAC) inhibitor) can effectively lower EV release by stimulating autophagic flux and thereby alleviate cellular stress (Figure 4). Of these compounds, valproic acid was the most effective in lowering EV release (Figure 4). This finding supports our hypothesis that cART interferes with lysosomal fusion, as valproic acid is a class I and II HDAC inhibitor that can target HDAC6, an enzyme responsible for the assembly of F-actin networks that facilitate fusion of the autophagosome with the lysosome [40,69]. This is supported by the data presented in Figure 5, which show a decrease in lysosome fusion using an RFP-GFP-tagged LC3B protein. The potential of cART to inhibit lysosomal fusion could have impacts beyond increased EV release. Interestingly, numerous studies have shown that astrocytes are primarily infected by HIV-1 through endocytosis, and that lysosomotropic agents such as chloroquine or Bafilomycin A1, which inhibit lysosome fusion, can increase HIV-1 infection [70,71,72]. It may therefore be possible that cART, especially therapies with reasonable BBB penetration, as utilized in this study, could potentially promote astrocyte infection by limiting the fusion of lysosome with endosomes that contain newly engulfed virus.

Our previous studies have shown that EVs containing TAR RNA can promote HIV-1 pathogenesis in recipient uninfected cells by eliciting an increase in proinflammatory cytokines and susceptibility to infection in recipient circulating immune cells [8,18,19]. Here, we show for the first time that EVs from HIV-1-infected myeloid cells can elicit changes in recipient astrocytes in terms of increased production and release of pro-inflammatory proteins (Figure 6). This could have broad implications for HIV-1 patients, including a reduction in the maintenance and support of neurons. The secretory phenotype elicited by EV treatment can propagate the inflammatory state of the CNS.

The high abundance of HIV-1 TAR RNA in patient biofluids and tissues, as well as the presence of TAR RNA in the majority of the tested patient materials, suggests that the impacts of TAR RNA could potentially have far-reaching effects. Our data confirm previous studies demonstrating that EV-associated HIV-1 TAR RNA can activate TLR3 in recipient cells, thereby contributing to chronic immune activation (Figure 7A,B). In general, the TLR3 signaling cascade can stimulate the transcription of Type I interferons via IRF3 and can activate the NF-κB pathway to produce pro-inflammatory cytokines. The data presented here suggest that the activation of TLR3 by EVs from infected myeloids may activate the IRF3 signaling cascade to induce subsequent production of proinflammatory cytokines, such as IL-6, from astrocytes (Figure 5 and Figure 6). However, EV-mediated activation could also activate the NF-κB pathway. NF-κB transcription factors are abundant within the CNS and can play both a protective and toxic role depending on the type of signaling [73]. While constitutive NF-κB signaling can provide neuroprotection, induction of NF-κB through various stimuli, such as TLR3 activation, has been found to contribute to neurological damage through several mechanisms. Induced NF-κB signaling can exert its effects on endothelial, neuronal, and glial cells. This induction leads to vascular inflammation and increased permeability of the BBB, neuronal cell death, and neurodegeneration and inflammation in glial cells, all of which are hallmarks of HIV-1-associated neurocognitive disorders [74,75,76,77]. Furthermore, NF-κB has been implicated in the induction of senescent phenotypes in a number of different models, and senescent astrocytes have been associated with chronic neurocognitive disorders such as Alzheimer’s disease [78,79,80]. Overall, astrocytes play a vital role in the support and maintenance of neurons as well as in the progression of CNS dysfunction in chronic diseases. Therefore, this potential EV-mediated TLR3 activation mechanism requires further investigation, specifically in the context of other CNS cell types or brain models.

To combat this mechanism, we have screened a panel of small-molecule TAR RNA-binding compounds for their ability inhibit the activation of TLR3 in recipient cells. Our findings show that three compounds, 103FA, 111FA, and Ral HCl, could selectively inhibit TLR3 activation by TAR RNA while maintaining innate immune function due to targeting of the ligand rather than the receptor (Figure 7C). Furthermore, the findings involving small TAR-binding molecules suggest that the changes in astrocytes are likely due to the activation of TLR3 by EV-associated TAR RNA (Figure 8). TLR3 is abundant in multiple CNS cell types, and some evidence suggests it is localized to the cell surface in astrocytes, potentially making them more vulnerable to TAR RNA-containing EVs (Figure 8) [44]. However, EVs possess a wide array of RNA cargo, including numerous cellular RNAs. As such, the possibility of cellular RNA contributions to TLR3 activation cannot be ruled out. In line with this, TLR3 expression is enhanced in the brains and spinal cords of individuals with chronic neuroinflammation, such as those with multiple sclerosis or Alzheimer’s disease [44,81,82], suggesting that these findings could potentially be applied to cellular RNAs encapsulated in EVs and their contributions to other chronic neurocognitive disorders. Docking of TAR-TLR3 revealed the interaction of TAR RNA with His539 of the C-terminus of TLR3 (Appendix A). His539 is a well-documented residue that is necessary for the recognition of dsRNA by TLR3, which is essential for dimerization and initiation of the signaling cascade [34,49,50,51]. Though the identified residues involved in the TAR-binding compound interaction are exclusive from those which mediate the interaction with TLR3, the TAR-binding compounds are effective in limiting TLR3 phosphorylation. These findings suggest that the mechanism through which these compounds inhibit TLR3 activation is non-competitive and likely induces a conformational change in TAR RNA.

In conclusion, despite antiretroviral therapy, HIV-1-infected macrophages have persistent viral transcription, which results in the accumulation of viral RNAs due to alteration in the autophagy pathway (Figure 9). This results in the release of increased viral RNAs and proteins incorporated into EVs, which can then affect neighboring cells such as astrocytes in the CNS. These EVs released from HIV-1-infected macrophages treated with cART can still induce TLR3 activation based on the presence of EV-associated double-stranded TAR RNA, which leads to subsequent proinflammatory cytokine production of IL-6. We show here that the TAR inhibitors 103FA, 111FA, and Ral HCl may lower EV-mediated TLR3 activation as well as downstream production of proinflammatory cytokines that can contribute to neuroinflammation.

## Figures and Tables

**Figure 1 viruses-16-00643-f001:**
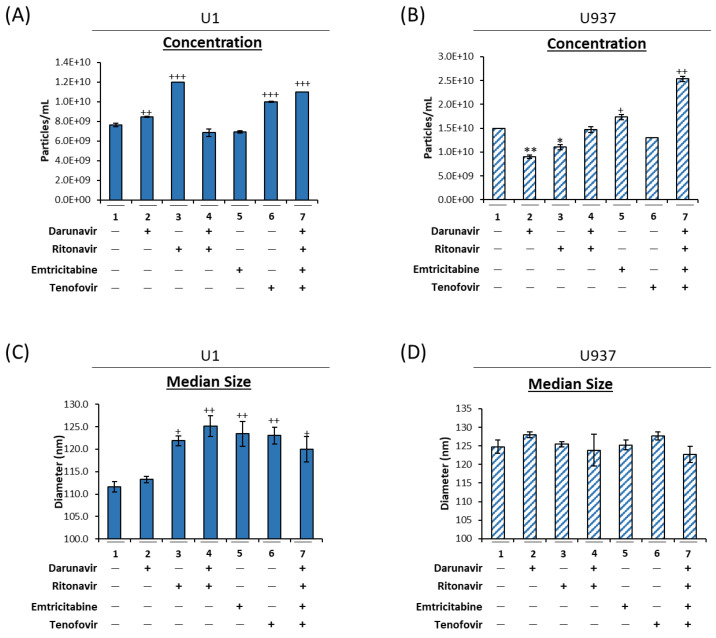
Antiretrovirals increase the release of EVs from HIV-1-infected monocytes. U1 (HIV-1-infected monocytes) and U937 (uninfected monocytes) cells were treated twice with darunavir (10 µM), ritonavir (5 µM), darunavir/ritonavir (10 µM/5 µM), emtricitabine (10 µM), tenofovir (10 µM), or a combined regimen (10 µM of emtricitabine, tenofovir, darunavir, and 5 µM of ritonavir) for 5 days in exosome-free media. Supernatants were analyzed using NTA to assess for drug-mediated changes in the released EV numbers in the treated U1 (**A**) and U937 (**B**) cells. An additional analysis was used to investigate the change in EV median diameter from U1 (**C**) and U937 (**D**) cells. Statistical significance was assessed using a two-tailed Student’s *t*-test comparing the treated samples (lanes 2–7) to an untreated control (lane 1). */+ *p* < 0.05, **/++ *p* < 0.01, +++ *p* < 0.001, in which */+ represents a significant decrease/increase, respectively.

**Figure 2 viruses-16-00643-f002:**
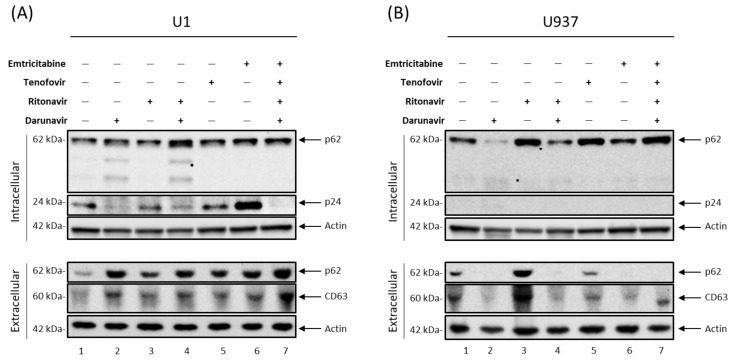
Antiretroviral drugs alter autophagy. U1 (HIV-1-infected monocytes; (**A**)) and U937 (uninfected monocytes; (**B**)) cells were treated twice with darunavir (10 µM), ritonavir (5 µM), darunavir/ritonavir (10 µM/5 µM), emtricitabine (10 µM), tenofovir (10 µM), or a combined regimen (10 µM of emtricitabine, tenofovir, darunavir, and 5 µM of ritonavir) for 5 days in exosome-free media. Following incubation, the cells and cell culture supernatant were harvested, and EVs were enriched from the supernatant using NT80/82. The intracellular material was analyzed via Western blot for the presence of the autophagy marker p62, HIV-1 viral proteins p24 (Gag protein), and Actin as a control. The extracellular (EV) material was analyzed via Western blot for the presence of p62, the EV marker CD63, and Actin as a control.

**Figure 3 viruses-16-00643-f003:**
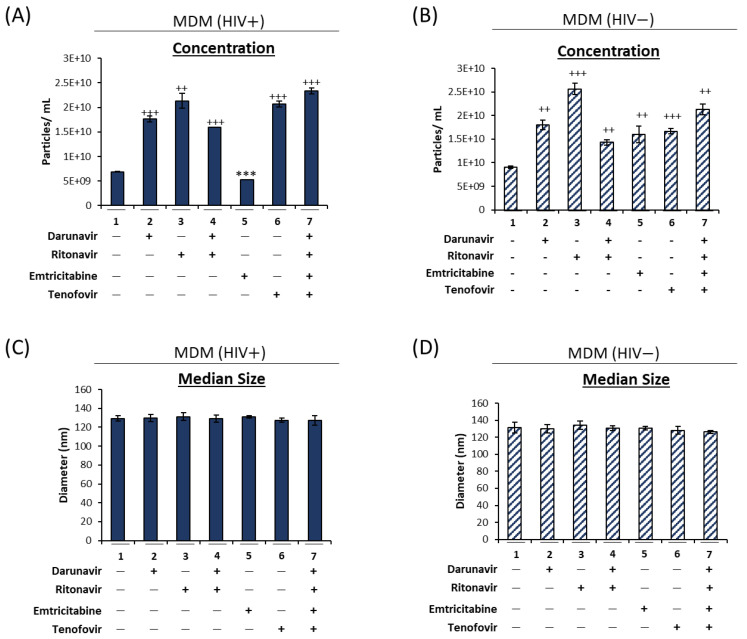
Antiretrovirals increase the release of EVs from HIV-1-infected monocyte-derived macrophages. U1 (HIV-1-infected monocytes) and U937 (uninfected monocytes) cells were differentiated into monocyte-derived macrophages (MDMs) using PMA (100 nM) over 7 days. Following differentiation, MDMs were treated twice with darunavir (10 µM), ritonavir (5 µM), darunavir/ritonavir (10 µM/5 µM), emtricitabine (10 µM), tenofovir (10 µM), or a combined regimen for 5 days in exosome-free media. The supernatants were analyzed using NTA to assess drug-mediated changes in the released EV numbers in the treated HIV+ MDMs (**A**) and HIV− MDMs (**B**). An additional analysis investigated the changes in the EV median diameter from HIV+ MDMs (**C**) and HIV− MDMs (**D**). Statistical significance was assessed using a two-tailed Student’s *t*-test comparing the treated samples (lanes 2–7) to an untreated control (lane 1). ++ *p* < 0.01, ***/+++ *p* < 0.001, in which */+ represents significant decreases/increases, respectively.

**Figure 4 viruses-16-00643-f004:**
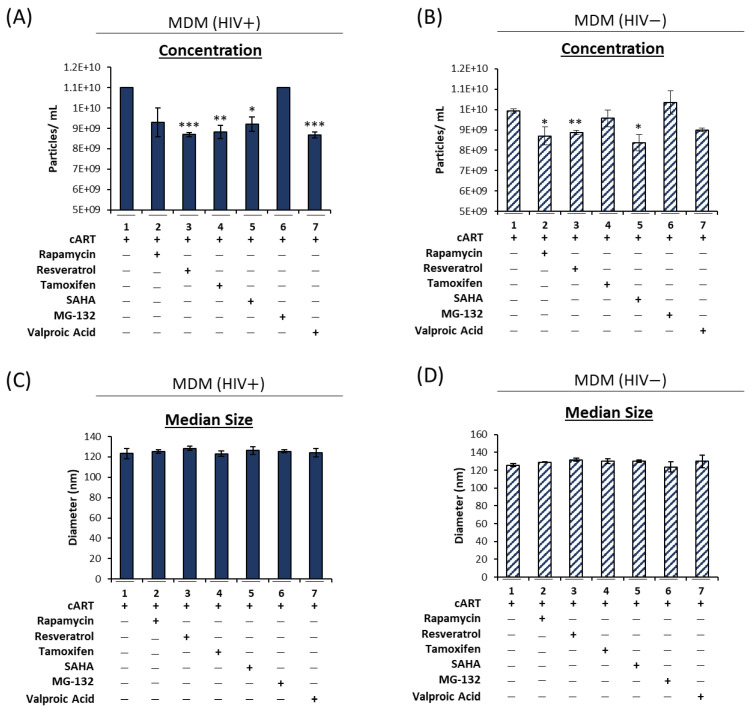
Autophagy-inducing compounds alleviate cART-mediated increases in EV release. U1 HIV-1-infected monocytes and U937 uninfected monocytes were differentiated into macrophages using PMA (100 nM). Post-differentiation MDMs were treated twice with cART (10 µM of emtricitabine, tenofovir, darunavir, and 5 µM of ritonavir) as well as autophagy-inducing compounds: rapamycin (100 nM), resveratrol (1 µM), tamoxifen (2.5 µM), SAHA (1 µM), MG-132 (50 nM), and valproic acid (60 µM) for 5 days. Following incubation, the HIV+ MDM (**A**) and HIV− MDM (**B**) supernatants were analyzed using NTA to assess changes in the number of EVs released. An additional analysis investigated changes in the EV median diameter from HIV+ MDM (**C**) and HIV− MDM (**D**). Statistical significance was assessed using a two-tailed Student’s *t*-test comparing the treated samples (lanes 2–7) to an untreated control (lane 1). * *p* < 0.05, ** *p* < 0.01, *** *p* < 0.001, in which * represents significant decreases.

**Figure 5 viruses-16-00643-f005:**
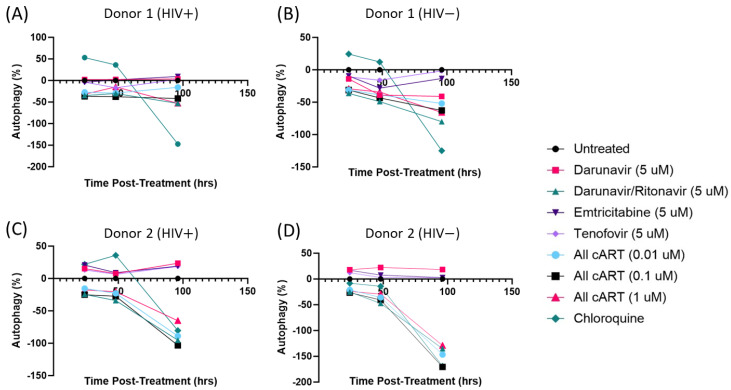
Treatment of primary macrophages with cART inhibits autophagosome fusion with the lysosome. Primary macrophages were infected with HIV-SF162 at an MOI of 1 and treated with antiretrovirals every 48 h. Autophagy was assessed using an RFP-GFP-LC3B tandem autophagy sensor. The experiments were performed in two independent donors, Donor 1 (**A**,**B**) and Donor 2 (**C**,**D**) +/− HIV-1 infection. Autophagy was measured at 24, 48, and 96 h post-treatment, and data points represent an average of 12 technical replicates.

**Figure 6 viruses-16-00643-f006:**
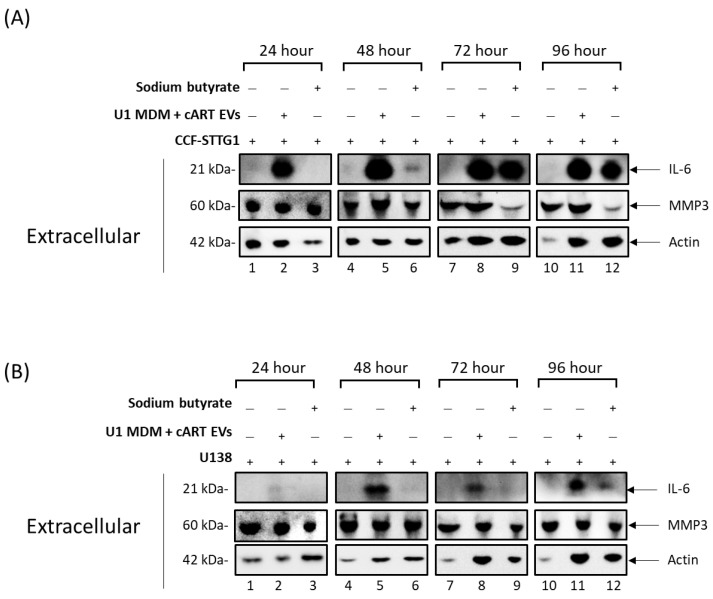
EVs released from HIV-1-infected MDMs undergoing cART induce secretion of proinflammatory cytokines from recipient astrocytes. HIV-1-infected U1 monocytic cells were cultured for 5 days. U1 monocytes were differentiated into MDMs using PMA (100 nM) and treated with ± cART cocktail (10 µM of emtricitabine, tenofovir, darunavir, and 5 µM of ritonavir) for 5 days. (**A**) CCF-STTG1 and (**B**) U138 cells (1 × 10^6^ each) were plated and treated with EVs released from MDMs treated with cART (1:10,000 cell to EV ratio) and sodium butyrate (1 mM). The supernatants were collected at 24, 48, 72, and 96 h and enriched for EVs using NT80/82 particles followed by SDS-PAGE and Western blot analysis for pro-inflammatory markers (IL-6 and MMP3) and Actin.

**Figure 7 viruses-16-00643-f007:**
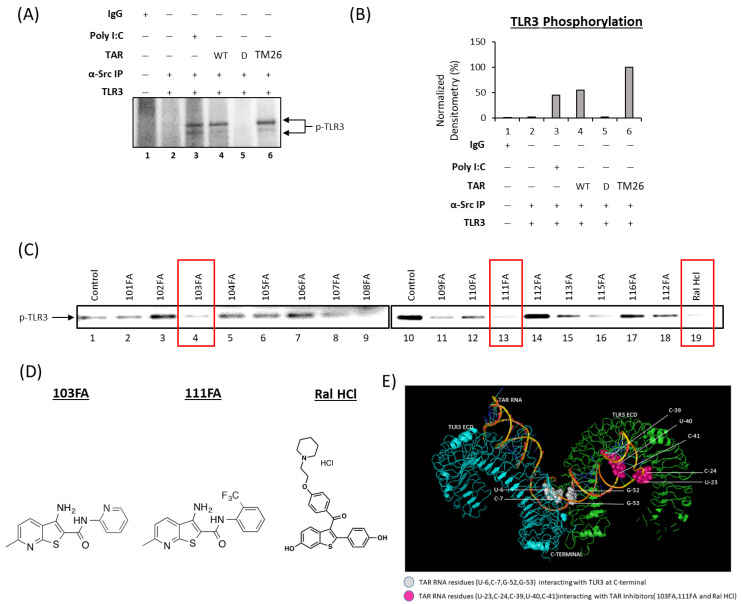
TAR-binding compounds mitigate activation of TLR3 by TAR RNA. (**A**) Confirmation of TLR3 activation by Src kinase in the presence of double-stranded TAR RNA was assessed using an in vitro kinase assay that utilized purified TLR3 protein (100 ng), purified Src kinase (10 ng), and TAR RNA (10 and 100 ng) run through SDS-PAGE. The blot was then imaged, post-stained, and dried via phosphor imaging. (**B**) Densitometry analysis of in vitro kinase assay was performed using ImageJ software (version 1.54h) to obtain raw densitometry counts relative to background. The counts are shown as relative expression of the protein relative to highest phosphorylation levels (lane 3; set to 100%). (**C**) Kinase assay using similar approach to panel A coupled with a panel of TAR inhibitors (1 µM), was performed for phosphorylation of TLR3 by Src kinase via Phosphorimaging with significant decrease highlighted with red boxes. (**D**) Chemical structures of TAR inhibitor compounds. (**E**) TAR-TLR3 docked structure showing the TLR3 ectodomain (green and cyan) form a dimer upon binding to TAR RNA (multicolor). The white spheres denote the TAR RNA residue positions (U6, C7, G52, and G53) that interact with TLR3, and the magenta spheres denote the TAR inhibitors (103FA, 111FA, and Ral HCl) with already published interaction positions (U23, C24, C39, U40, C41) on TAR RNA (41). This image was visualized using PyMol (50).

**Figure 8 viruses-16-00643-f008:**
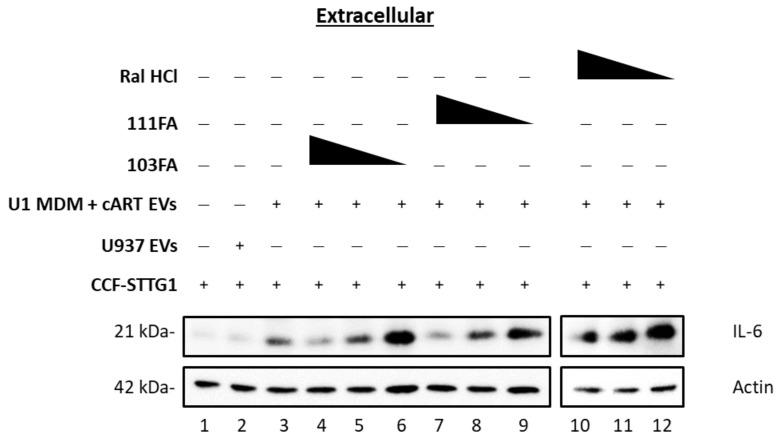
HIV-1 TAR inhibitors lower proinflammatory cytokine production in astrocytes. U937 and U1 cells were cultured for 5 days. U1 monocytes were differentiated into MDMs using PMA (100 nM) and treated ± cART cocktail (10 µM of darunavir, emtricitabine, tenofovir, and 5 µM of ritonavir) for 5 days. EVs were isolated using ultracentrifugation and assessed for concentration and particle size using ZetaView NTA to equilibrate EV concentrations. EVs were used to treat astrocytes (CCF-STTG1; 1 × 10^6^) at a 1:10,000 concentration (cell/EVs ratio). CCF-STTG1 cells were treated with ± EVs from U937 cells, as well as U1 MDMs that were treated with cART (10 µM of darunavir, emtricitabine, tenofovir, and 5 µM of ritonavir) at a 1:10,000 concentration (cell/EVs ratio), and ±titration of TAR inhibitors (100, 10, and 1 nM; 103FA, 111FA, and Ral HCL) for 3 days. EVs from supernatants were enriched using NT80/82 particles and analyzed for the presence of IL-6 and Actin using Western blot.

**Figure 9 viruses-16-00643-f009:**
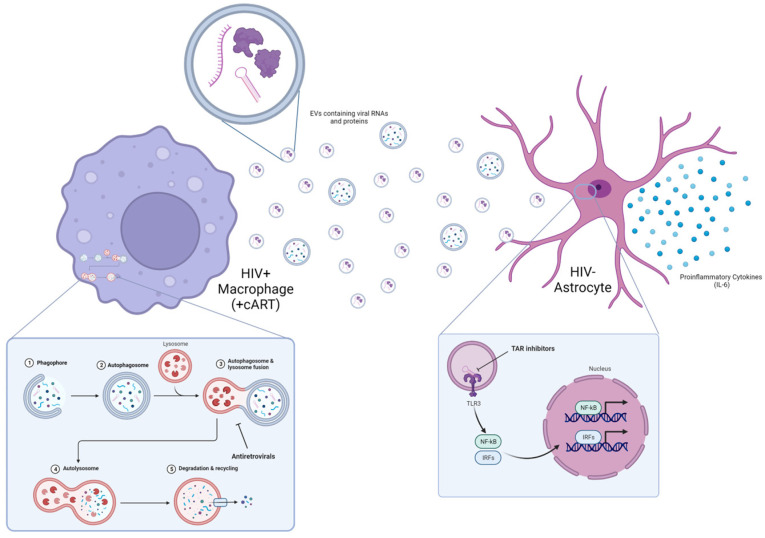
Persistent viral transcription in HIV-1 infected myeloid cells contributes to neuroinflammation. Continued viral transcription in myeloid lineage cells yields an accumulation of viral transcripts that is exacerbated by antiretroviral treatment via alternations in the autophagy pathway. To alleviate stress, cells release viral products from the cell in extracellular vesicles that can trigger the innate immune response via TLR3, causing an increase in the release of proinflammatory cytokines.

**Table 1 viruses-16-00643-t001:** PATCHDOCK analysis of TLR3 interaction with TAR RNA.

Rank	Geometric Shape Complementarity Score	Interface Area	ACE (kcal/mol)
**1**	18,466	4013.50	−1492.78
**2**	17,732	3228.50	−1197.85
**3**	17,870	3453.60	−1034.96
**4**	19,060	3353.60	−1009.48
**5**	18,980	3371.30	−967.94
**6**	18,072	3202.80	−638.25
**7**	20,688	3171.10	−624.67
**8**	18,246	3080.00	−531.61

## Data Availability

The data are available upon request. Please contact Fatah Kashanchi at fkashanc@gmu.edu.

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
