# Peer review of "Autophagy Deregulation in HIV-1-Infected Cells Increases Extracellular Vesicle Release and Contributes to TLR3 Activation"

_viruses, 2024, doi:10.3390/v16040643_

Round 1

Reviewer 1 Report

Comments and Suggestions for Authors

In this study, DeMarino and colleagues show that combined antiretroviral therapy (cART) used for combating HIV-1 infection, could contribute to autophagy deregulation in infected cells and increase in extravesicular vesicles (EVs) released from infected cells that could contribute to CNS pathogensis. The results presented here could be important to better understand this topic, but a few additions are needed to increase the impact of this study and support the authors conclusion.

1. What is the titer of HIV-1 released by U1 cells without and with cART treatment? It is an important information that should be provided.

2. Figure 1. A. Lane 4 (Darunavir+Ritonavir) and Lane 5 (Emtricitabine) show lower EVs production. Is this significant? Why are they lower?

3. Fig. 1. What is the reason for which U937 (uninfected monocytes) produce more and larger EVs than U1 HIV-1 infected? Is this difference significant?

4. What is the number of EVs released by primary macrophages infected vs uninfected, treated with cART vs untreated?

5. For TLR3 activation by TAR RNA using kinase assay, a control (scrambled) RNA should be used for the specificity of activation.

6. Do the EVs released from HIV-1 infected primary macrophages treated with cART induce similar inflammatory response as the EVs from HIV-1 infected U1 cells?

Reviewer 2 Report

Comments and Suggestions for Authors

Dear Authors,

In this paper written by DeMarino at al, authors demonstrated that EV release was affected by cART, HIV infection and cell differentiation. Furthermore, the paper strongly suggested that TAR in the released EVs induces IL6 from astrocytes through TLR3 activation by Src. The paper provided very interesting observations and opinions in this field, and it fits well to the scope of the journal. I am pleased to read the paper and positive to recommend the publication but believe that revision of the manuscript benefits the readers more. Some points to consider in subsequent versions are shown below.

1.            References for U1 and U937 should be provided.

2.            The nanotracking analysis with the U937 cells infected with HIV-1 could be added. The experiments could exclude potential genetic differences in the cells against the cART drugs and will provide better understanding about EV release from the monocytes in general.  

3.            Line 321-323 showed “Data of Figure 2A show an increase in the intracellular p62 levels when HIV-1 infected monocytes were treated with antiretrovirals”. Does the Supplementary Figure 2A show analysis of the pictures of Figure 2A (U1 cells)? Authors can add error bars in Suppl. Figure 2 and show statistical analysis to support the statement “increase”.

4.            Authors could consider addition of a bar graph that analyzed the Figure 2B.

5.            Figure 3 provided the interesting observations in EV release in the MDMs. Did HIV infection affect the differentiation by PMA? How did authors assess cell differentiation after PMA treatment?

6.            Similar to Figure 2, Figure 3 could need the data with the U937 cells infected with HIV-1.  

7.            Lines 422-423, pharmacological approach to induce autophagy decreased the number of EVs released from the HIV-1 infected cells (figure 4). Authors should consider confirmation of autophagy flux (e.g. p62, LC3-II/LC3-I or Actine) in MDM with the chemicals.

Reviewer 3 Report

Comments and Suggestions for Authors

De Marino and coll have undertaken an interesting study to demonstrate that antiretroviral therapy exacerbate packaging of viral products into extracellular vesicles (EVs), (EV), favoring the production from astrocytes of inflammatory cytokines in the CNS. Therefore, the study seems to be focused on CNS inflammation in HIV treated patients.

I would like to ask for some clarifications.

Material and Methods.

The authors should clarify more in  details the isolation /characterization of EV.

Results: The author should explain whether  they have  assayed the cytotoxicity of antiretroviral used toward U937 cells

The authors  should indicate why they have chosen to use antiretrovirals at the concentration tested. Did they perform a dose-effect assay?.

They have used U937 cells for the reported experiments, but it is interesting to know whether they have infected monocytes/macrophages from healthy donors  and which results they have obtained , using the identical culture conditions than in U937 cells

Discussion

The authors sustain that the autophagy deregulation might be one of the cause of CNS inflammation In HIV infected individuals  but I would like they comment  on the fact that HIV patients can survive thank to ART therapy and how it is possible to adjust the need of ART with the limitation of side effects. A chance  could be to combine ART with  TLR3 inhibitors?. The authors should more clearly comment how the therapy of HIV could change to the light of taking into the account dysregulation of mechanism of cell defense such autophagy, associated to ART therapy. This is a highly key aspect to further progress in the therapy of HIV.

Reviewer 4 Report

Comments and Suggestions for Authors

The paper investigated the effects of antiretroviral therapy on extracellular vesicle (EV) release from HIV-1 infected cells and autophagy deregulation in the HIV-1 infected cells, and their connection. The autophagy deregulation in HIV-1 infected cells increases EV release; and the released EVs contribute to TLR3 activation. The released EVs contain double stranded TAR RNA from HIV-1, which leads to production of IL-6. They further show that treated with TAR inhibitors, 103FA, 111FA, and Ral HCl, lower EV-mediated TLR3 activation (Figure 7B), as well as downstream production of IL-6 (Figure 8).

However there are some minor issues need to be addressed:

1.        In the result section (section 3), there is no subtitles for section 3.2 and 3.3

2.        Figure 1 B lane 6: a. Is there an error bar for the data of lane 6?  In the text (line 285-286), it reads: “Conversely, in the absence of HIV-1 infection, Darunavir (lane 2), Ritonavir (lane 3), and Tenofovir (lane 6) alone decreased EV production (Figure 1B)”.  However in Figure 1 B lane 6 has no indication label on the column. Is the decrease for Tenofovir (lane 6) alone significant compared to the control?  Is that the“ + ” also mean significant different? Please add description in the figure legend (wherever is relevant).  

3.        Figure 1C lane 7, Please check if the data is significant. In the text (line 289-290) :“Data of Figure 1C show an increase in the median diameter size of EVs released from HIV-1 infected monocytes following treatment with every tested cART drug (lanes 2-7).”

4.        Line 391:These findings are distinctively different than those observed for monocytes (Figure 3B),” please check if it means Figure 1B.

5.        Figure 4B: Please check the data for lane 3, 5 and 7, are they significant compared to the control? And the description (line 426-428): “addition of autophagy inducing compounds caused little to 426 no change in the number of EVs released from uninfected cells, with the exception of Rapamycin, which caused a 14% reduction (Figure 4B). 

6.        Figure 5 only provide data from 2 donors, is that possible to provide data from more donors from statistic point of view?

7.        Figure 7 A-B: is that possible for the authors to provide another blot image for Extracellular 24 h MMP?

8.        Line 550-551: “Figure 6E shows the top ranked solution visualized by PyMol (50).” Is that Figure 7E?

9.        Line 556: “G53 interact with TLR3 (Figure 7D).” Is that Figure 7E?

10.     Line 560 and 563, please check “data of Figure 7 indicate that another TLR3 C-terminus residue, Asn515, a known 560 RNA-binding residue which has not been shown to be required for TLR3 activation, also 561 binds TAR RNA [32]. Along the same line, TLR3 residues Asn541 and Leu 595 were found 562 to interact with TAR RNA at C7 and G53 positions, respectively (Figure 7).” Is that Supplementary Figure 8?

11.     Line 582-583: “NTA for EV concentration (Supplementary Figure 4B). Median size similarities were also 582 confirmed by NTA (Supplementary Figure 4C). Normalized EV preps at a ratio of”. Please check the figure citation. There is no Supplementary Figure 4C.

12.     There is no description for Supplementary Figure 9. Please insert the description in the text.

Comments on the Quality of English Language

see above

Round 2

Reviewer 2 Report

Comments and Suggestions for Authors

Dear Authors,

The revised manuscript responded to the questions I raised and provided the detailed information of the methods with the additional data. I appreciate the responses and improvement of the manuscript. I highly recommend to publish the paper in Viruses.  

Reviewer 3 Report

Comments and Suggestions for Authors

The manuscript has been properly improved and it can be accepted in the present form

Comments on the Quality of English Language

The manuscript deserves publication since the authors have replied correctly and exhaustively to the referee's questions/comments